# PseDet: Revisiting the Power of Pseudo Label in Incremental Object Detection

**Qiuchen Wang**[1], **Zehui Chen**[1], **Chenhongyi Yang**[2], **Jiaming Liu**[3], **Zhenyu Li**[4], **Feng Zhao**[1]*

[1]MoE Key Laboratory of Brain-inspired Intelligent Perception and Cognition, USTC
[2]University of Edinburgh [3]Peking University [4]King Abdullah University of Science and Technology
`qiuchenwang@mail.ustc.edu.cn`
`fzhao956@ustc.edu.cn`

## Abstract

Incremental Objection Detection (IOD) facilitates the expansion of the usage scope of object detectors without forgetting previously acquired knowledge. Current approaches mostly adopt response-level knowledge distillation to overcome forgetting issues, by conducting implicit memory replay from the teacher model on new training data. However, this indirect learning paradigm does not fully leverage the knowledge generated by the teacher model. In this paper, we dive deeper into the mechanism of pseudo-labeling in incremental object detection by investigating three critical problems: (a) the upper bound quality of the pseudo labels is greatly limited by the previous model, (b) fixed score thresholds for label filtering, without considering the distribution across categories, and (c) the confidence score generated by the model does not well reflect the quality of the localization. Based on these observations, we propose a simple yet effective pseudo-labeling continual object detection framework, namely **PseDet**. Specifically, we introduce the spatio-temporal enhancement module to alleviate the negative effects when learning noisy data from the previous model. Considering the score distribution divergence across different classes, we propose the Categorical Adaptive Label Selector with a simple mathematical prior and fast K-Means pre-computation to dynamically determine the class-wise filtering threshold. In order to align the label score with the localization quality of the pseudo labels, we project the score through non-linear mapping to calibrate the distribution and integrate it into the new-step supervision. Extensive experiments on the competitive COCO benchmarks demonstrate the effectiveness and generalization of PseDet. Notably, it achieves 43.5+/41.2+ mAP under the 1/4-step incremental settings, achieving new state-of-the-art performance. Code is available at https://github.com/wang-qiuchen/PseDet.

## 1 Introduction

Object detection is a fundamental and important task with various applications (Li et al., 2020; Tian et al., 2019; Zhang et al., 2020b; Chen et al., 2021). However, static detectors cannot well fit the needs of real-world scenarios since the changing environments and extra demands require the model to seamlessly adapt to new class detection with updated training data (Joseph et al., 2021; Caccia et al., 2021). Such a problem, which refers to incremental object detection (IOD), denotes the ability of object detectors to acquire new knowledge without forgetting (Liu et al., 2023b;a). Concretely, training samples for different categories are observed at varied training steps, and the detectors are restricted from accessing the ground truth labels in the past steps.

Previous approaches mainly adopt knowledge distillation (KD) (Chen et al., 2017; Hinton et al., 2015) to overcome the catastrophic forgetting problem. By distilling from the intermediate products generated by the teacher model, it provides direct supervision to enforce the new model to be close to its former one. This imposes an implicit regularization to the update of the training model. Such a distillation framework can also be viewed as a special case of pseudo-labeling, where the distillation targets are actually treated as pseudo-labels during new-step training.

---

*Corresponding author.

Despite being effective, it does not fully anticipate the potential benefits of pseudo-labels. Specifically, we observe three critical issues in current paradigms: (1) the upper bound quality of distillation supervision is greatly limited by the previous (teacher) model. Due to the learning ability of different detectors, the localization quality generated by the previous model is far from the ground truth. Directly enforcing the target model to mimic the knowledge from these noisy (mixed-quality) data can confuse the learning process of the model and even deteriorate the performance. (2) object detectors tend to exhibit score biases across different categories (Figure 3) due to the data distribution and learning difficulty of the training labels. However, current works mostly set hard thresholds for all classes for label filtering, *e.g.,* 0.3, which overlooks the score divergence at the category level, and (3) in order to fully leverage the information generated from the pseudo labels, we carefully examine the relationship between confidence score and localization quality. Though the confidence score is viewed as the metric to reflect the quality of detected bounding boxes, we empirically find these two variables do not demonstrate linear correlations. Therefore, simply taking the confidence score as the label quality for new-step training may introduce extra noise for the model.

Based on the above observations, we propose a simple yet effective pseudo-labeling continual object detection framework, namely **PseDet**, which fully leverages the intermediate labels for effective model regularization. To eliminate the noisy information from the immature previous model, we introduce the Spatio-Temporal Enhancement Module, which enhances the label quality along the spatial and temporal dimensions. By augmenting inputs at different input scales and different steps, we seamlessly improve the quality of the generated labels with the multi-group ensemble. In terms of the score distribution divergence across different classes, we propose the Categorical Adaptive Label Selector to dynamically determine the threshold on different categories with a simple mathematical prior and fast K-Means pre-computation. It removes the massive human efforts to manually select the filtering threshold for each class and greatly ameliorates the final label quality. Besides, to align the confidence score and localization quality of the generated labels, we project the score through a non-linear mapping function to calibrate the distribution and seamlessly integrate it into the new-step training. Thanks to these enhancements on the pseudo-labeling, PseDet achieves 43.5+/41.2+ mAP on the challenging COCO dataset under 1/4 steps training settings, surpassing previous state-of-the-art by a large margin.

Our contributions are summarized in three-fold:

- We revisit the strategy of pseudo-labeling on incremental object detection and identify three critical problems that hinder it from achieving competitive performance.
- Based on the above findings, we propose PseDet, a simple yet effective pseudo-labeling framework for incremental object detection, which consists of three key components: spatiotemporal enhancement module, categorical adaptive label selector, and confidence score calibration supervision.
- Extensive experiments conducted on the MS COCO dataset with various incremental settings validate the effectiveness and generalization of our approach. Notably, PseDet outperforms previous methods by 4∼17 mAP on different learning settings, achieving new state-of-the-art in incremental object detection.

## 2 RELATED WORKS

### 2.1 INCREMENTAL LEARNING

Incremental learning has been studied by the machine learning community for a long time (Schlimmer & Granger, 1986; Masana et al., 2022; Joshi & Kulkarni, 2012), with the main goal of alleviating the catastrophic forgetting problem (Kemker et al., 2018; Goodfellow et al., 2013; Hayes et al., 2020). Generally speaking, there are four classes of approaches. Firstly, memory-based approaches (Chaudhry et al., 2018; Zhang et al., 2020a; Lopez-Paz & Ranzato, 2017; Caccia et al., 2021) use a small-sized cache to store information from previous tasks, which are replayed when training the model on new tasks. Secondly, during the current task training, regularization-based methods (Kirkpatrick et al., 2017; Zenke et al., 2017; Li & Hoiem, 2017) regularize the updating of those model parameters that are important to previous tasks. For example, EWC (Kirkpatrick et al., 2017) measures the parameter importance using Fisher information matrices, whose computation efficiency is improved in the subsequent work SI (Zenke et al., 2017). Thirdly, parameter

isolation-based methods (Rusu et al., 2016; Yoon et al., 2019; Fernando et al., 2017; Kang et al., 2022), e.g., PathNet (Fernando et al., 2017) aims to separate the model parameters of each task by learning task-dependent sub-networks, so that they will not be able to affect each other. Finally, the knowledge distillation-based approaches (Li & Hoiem, 2017; Rebuffi et al., 2017; Wu et al., 2019), e.g., LwF (Li & Hoiem, 2017), explicitly inject task-dependent information of previous tasks through knowledge distillation. For example, in iCaRL (Rebuffi et al., 2017), knowledge distillation is used to alleviate the excessive deterioration problem.

## 2.2 Incremental Object Detection

While there is a significant amount of research focusing on incremental learning for image classification, solving the incremental learning for object detection (Peng et al., 2021; Perez-Rua et al., 2020; Li et al., 2019; Joseph et al., 2021; Feng et al., 2022) is a non-trivial task. (Liu et al., 2020) first proposes a weight consolidation approach by applying EWC to two-stage detectors like Faster R-CNN. MVCD (Yang et al., 2022) introduces feature and response distillation by splitting channels and spatial features for model regularization. Elastic Response Distillation is proposed in (Feng et al., 2022) to conduct knowledge distillation on detector classification and regression branches. CL-DETR (Liu et al., 2023a) explores the incremental detection setting in the scope of detection transformer. SDDGR (Kim et al., 2024) leverages a generative model based on stable diffusion to mitigate catastrophic forgetting. Besides, (Joseph et al., 2021) leverages open-world approaches to integrate incremental learning and open-set object detection.

## 3 PseDet

In this section, we first provide the problem formulation on incremental object detection (§ 3.1). Then we briefly introduce the overall framework of PseDet (§ 3.2), which consists of three key components, Spatio-Temporal Enhancement Module (§ 3.3), Categorical Adaptive Label Selector (§ 3.4), and Confidence Score Calibration Supervision (§ 3.5).

## 3.1 Problem Formulation

Incremental object detection (IOD) aims to detect interested objects across class domains in multiple steps, where we assume that there are $N$ steps. Let $\mathcal{D}$ denotes a complete dataset comprising images $x$ and their corresponding annotations $y$, which can be represented as $\{x, y | (x, y) \in \mathcal{D}\}$. Meanwhile, let $\mathcal{C}$ represent the category of the annotated objects in $\mathcal{D}$, and we partition the dataset into $N$ subsets within the class domain $\mathcal{C} = \mathcal{C}^1 \cup \cdots \cup \mathcal{C}^N$. In step $i$, the model can only access the labels belonging to category $\mathcal{C}^i$, but it cannot access previously learned categories $\mathcal{C}^{1:i-1}$, even if objects of those categories exist in the images. The model needs to maintain the ability to detect class $\mathcal{C}^{1:i-1}$ while learning to detect new ones $\mathcal{C}^i$, as each class is only learned once.

## 3.2 Overall Framework

The overall framework of our approach is shown in Figure 1. During each step of training, PseDet first prompts the previous model to generate improved pseudo labels with the proposed **Spatio-Temporal Enhancement** module, by attenuating noise at its source through label fusion in both the spatial and temporal domains. Then, the **Categorical Adaptive Label Selector** dynamically adjusts thresholds for each class within every mini-batch, automating the whole filtering process to assure the quality of pseudo labels across different categories. Finally, when computing the loss $\mathcal{L}_{cls}$ on the target model, we introduce the **Confidence Score Calibration Supervision** to align the confidence score with the localization quality through non-linear mapping and seamlessly integrate it into the supervision process.

## 3.3 Spatio-Temporal Enhancement Module

In most incremental object detection frameworks, the detection performance of learned categories gradually declines as the learning progresses step by step. This is possibly due to the reorganization of the parameter space, which is caused by the limited model capacity as new knowledge is learned.

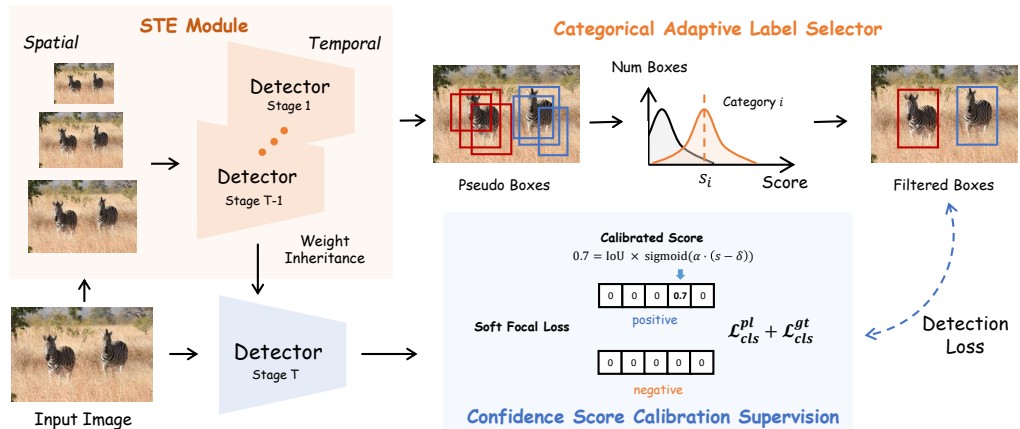

Figure 1: **The overall framework of PseDet for incremental object detection**. We introduce three critical components to maximize the power of pseudo-labels in IOD: Spatio-Temporal Enhancement module, Categorical Adaptive Label Selector, and Confidence Score Calibration Supervision.

However, we assume that such a decline can also be attributed to noise interference during the knowledge transfer process. Maximizing the signal-to-noise ratio of the information source and minimizing noise in the transmission process is crucial for improving the accuracy of information transmission. Inspired by this, we enhance pseudo-labels at the spatial domain to reduce the discrepancy when inferring pseudo-labels from the old model. Moreover, in multi-step scenarios, we conduct fusion in the temporal domain to prevent the accumulation and magnification of noise across steps.

Current approaches enhance supervision by leveraging zero-shot learning methods to generate labels or by simply replaying labels. To avoid introducing more complex networks and operations, we employ a straightforward strategy by applying a series of spatial transformations to the inputs. Inputs processed through different augmentation techniques can aid the model in detecting objects of varying types and sizes, thus improving the robustness of the model. Formally, we define an augmentation transformation set $\mathcal{A}$, which encompasses transformations applied to image tensors in the spatial domain. For input image $x \in \mathcal{D}$ transformed via $\mathcal{A}_i$, we use the old model $\Phi_{old}$ to predict objects that belong to the old categories:

$$y_i = \Phi_{old}(\mathcal{A}_i(x)), \tag{1}$$

where $y_i$ is the prediction of the model $\Phi_{old}$ after transformation $\mathcal{A}_i$. Note that all these augmentation operators are invertible. Once the predictions are obtained, we simply reverse them back to the original space and perform unified fusion operations:

$$\mathcal{P} = \mathcal{F}(A_1^{-1}(y_1), \cdots, A_i^{-1}(y_i)), \tag{2}$$

where $\mathcal{P}$ is the set of pseudo-labels passed to the next step, $\mathcal{A}_i^{-1}$ denotes the reverse transformation of $\mathcal{A}_i$, for inputs that are horizontally flipped, $A^{-1}$ re-flips the model's output horizontally; for scaled inputs, it scales them back to their original size. $\mathcal{F}$ represents spatial fusion, typically using NMS with an IoU threshold of 0.4.

The quality of pseudo-labels generated by the teacher model decreases as the incremental learning progresses. In the context of multi-step incremental learning, the knowledge acquired earlier is contaminated by accumulated noise more severely as the training continues. This contamination can potentially deteriorate the model's ability to effectively learn new classes. Therefore, we introduce the temporal domain enhancement to minimize the model's susceptibility to noise interference. Suppose the model can access ground truth labels of categories $\mathcal{C}^i$ and pseudo-labels of categories $\mathcal{C}^{i-1}$ at step $i$, we collect the optimal pseudo-labels $\mathcal{P}$ in the temporal domain:

$$\mathcal{P} = \mathcal{P}^1 \cup \cdots \cup \mathcal{P}^{i-1}, \tag{3}$$

where $\mathcal{P}^{1:i-1}$ represents the pseudo-labels generated by the model $\Phi_{1:i-1}$, containing the set of categories $\mathcal{C}^{1:i-1}$. By combining the spatial and temporal enhancement, our PseDet greatly alleviates the noise influence from the old label and greatly improves the label quality.

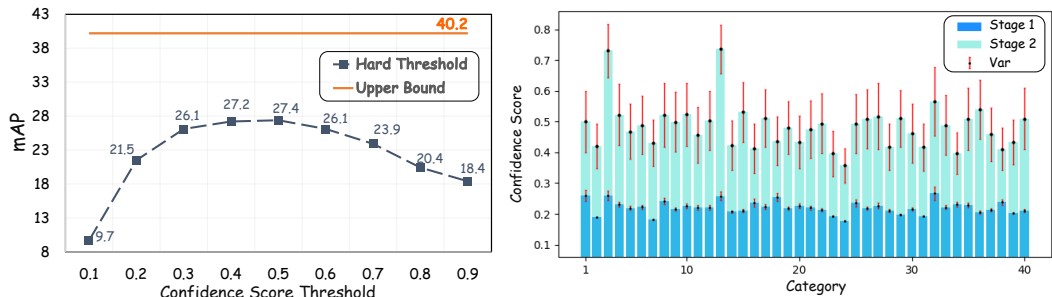

Figure 2: Experimental results with different hard thresholds, showing the impact of filter on performance.

Figure 3: The class-wise confidence score distributions in the 40-40 experiment, show its variability across different classes and steps.

## 3.4 CATEGORICAL ADAPTIVE LABEL SELECTOR

In IOD tasks, the key reason for catastrophic forgetting by the detector is the shift between the new labels and the previously learned ones (Liu et al., 2023b). While images contain both old and new classes, annotations only contain labels for objects of the current class, causing the model to consider the old classes as background during the fitting process to the current dataset of this step.

Pseudo-labeling provides an effective solution to transfer such knowledge across different steps. Simply applying different thresholds to filter labels based on confidence score leads to fluctuated performance depending on the threshold selection. As shown in Figure 2, a common issue in pseudo-labeling methods is that the performance of the detector is highly sensitive to the threshold. When the threshold is too small, a large amount of noise can mislead the detector's training. Conversely, when the threshold is too large, many valid pseudo-labels may be discarded, leading to significant forgetting. Not only that, as shown in Figure 3, we observe that the score distribution of pseudo-labels changes with each step in continual learning. It is unrealistic to manually determine the threshold for each class, especially under the incremental learning setting. Therefore, it is non-trivial to automatically determine the score threshold for label quality filtering.

Interestingly, we observe a distinct clustering of confidence scores for noisy and high-quality boxes, primarily in the low-confidence region, with evident class-specific characteristics. Based on the varying distributions of true positives (TP) and false positives (FP), we employ K-Means to fit these distributions for different classes, dynamically determining class-wise thresholds.

---

**Algorithm 1** Pseudo label selection in stage $i$

---

**Input:** candidate pseudo label set $\mathcal{P}_{input}$; k-means input queue length $N$.
**Output:** selected pseudo label $\mathcal{P}_{output}$
    Initialize the confidence score queue $\mathcal{Q}^c \leftarrow \{\}$ for each category $c \in \mathcal{C}^{1:i-1}$
    **for** $(\mathcal{S}, \mathcal{C}) \in \mathcal{P}_{input}$ **do**
        Add the confidence score to the queue $\mathcal{S}^c \rightarrow \mathcal{Q}^c$
    **end for**
    **for** $c \in \mathcal{C}^{1:i-1}$ **do**
        K-means$(\mathcal{Q}^c) \rightarrow \mathcal{D}_T^c, \mathcal{D}_F^c$
    **end for**
  $\mathcal{P}_{output} = \mathcal{D}_T^1 \cup \cdots \cup \mathcal{D}_T^{i-1}$.

---

Algorithm 1 provides a detailed demonstration of dynamic class-wise pseudo-label filtering process. As the distribution of confidence varies across classes, we initialize a queue $\mathcal{Q}^c$ for each category $\mathcal{C}^i$. We first enqueue the samples $\mathcal{S}^c$ of this batch, while the sample at the front of the queue will be dequeued. For each queue, we apply K-Means to fit the true and false distributions. The final output is the union of positive samples across all categories. Our method effectively filters noise by combining short-term memory with statistical methods, achieving high performance.

### 3.5 Confidence Score Calibration Supervision

Different from the carefully human-annotated ground truth obtained at the current stage, pseudo-labels are inferred from the previous detector, containing a significant amount of unexpected noise. It not only leads to forgetting but also interferes with the learning of new classes at the current step. Previous works tend to filter out low-confidence noise directly to retain accurate pseudo-labels, which leads to significant knowledge forgetting. As shown in Figure 2, using a higher threshold directly to filter noise $\mathcal{P} = \{(s^c, c)|s^c > s_{thres}\}$ results in a significant performance drop.

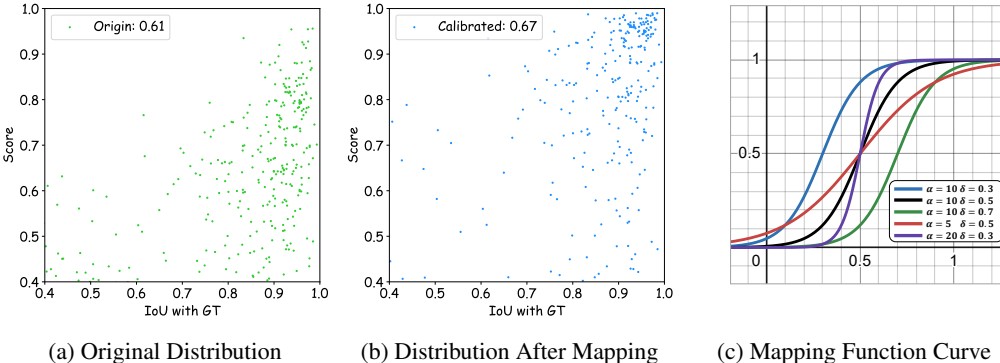

| (a) Original Distribution | (b) Distribution After Mapping | (c) Mapping Function Curve |

Figure 4: **Distribution of IoU-Score for pseudo-labels before and after mapping, along with the plot of the mapping function**. In the top left corner of the distribution plot is the Pearson Correlation Coefficient (higher is better), which measures the strength of the correlation between IOU and score. From (a) and (b), it can be observed that after mapping, labels with high IoU are concentrated in the high-score region compared to before mapping, benefiting the model's attention to high-quality pseudo-labels.

Recent work (Jiang et al., 2018) suggests that confidence scores can indicate label accuracy, implying higher attention should be given to samples with higher scores during model training. However, our findings show a non-linear relationship between label quality and confidence scores. Therefore, we map the confidence score $s$ to a quality coefficient $q$, which better represents label accuracy. As shown in Figure 4a, the scatter plot of IoU versus $s$ is not linear around the line $y = x$. In the interval $s = [0.5, 1]$, bounding box quality improves significantly, while in $s = [0, 0.5]$, accuracy is generally lower. The sigmoid function is suitable for mapping $q$ and $s$, with translation and scaling to ensure it operates within [0,1]:

$$q(s) = sigmoid(\alpha \cdot (s - \delta)) = \frac{1}{1 + e^{-\alpha \cdot (s^c - \delta)}}, \tag{4}$$

where $\alpha$ is the scaling coefficient, $\delta$ is the translation distance. We set $\alpha = 10$ and $\delta = 0.5$ in the implementation. Figure 4c illustrates the graph of the mapping function, and Figure 4b illustrates the distribution after mapping.

Instead of directly assigning pseudo labels and ground truth labels to each prediction as 0 or 1, we use the $q_{pl}$ and the IoU $\tau$ between predictions and their matched labels to soften the categorical labels. Unlike GFL (Li et al., 2020), which assigns labels based on the IoU between network predictions and labels, we consider the quality of labels as an important factor in our approach, where we transform Quality Focal Loss into a form suitable for continuous label representations in IOD:

$$\hat{y}_{pl} = \tau \cdot q_{pl}, \tag{5}$$

where $\hat{y}_{pl}$ is the joint quality, $q$ is the quality of the pseudo label, and $\tau$ is the IoU between the label and its assigned prediction. The class loss for the pseudo labels is:

$$\mathcal{L}_{cls}^{pl} = -|y - \hat{y}_{pl}|^\beta ((1 - y)\log(1 - \hat{y}_{pl}) + y\log(\hat{y}_{pl})), \tag{6}$$

where $y$ is the predictions of the current step model, $\beta$ is set to 2. For the ground truth, which is universally recognized as high quality $q_{gt} = 1$, allowing us to uniformly represent $\hat{y}_{pl}$ and $\hat{y}_{gt}$ with $\hat{y}$:

$$\hat{y}_{gt} = \tau \cdot q_{gt} = \tau \cdot 1, \tag{7}$$

Table 1: **Incremental results on COCO benchmark under the one-step setting**. Most experimental results are borrowed from SDDGR (Kim et al., 2024). $AP$, $AP_{50}$, and $AP_{75}$ reflect the overall performance (higher is better) of the model after one step of learning. AbsGap (lower is better) and RelGap (lower is better) represents the absolute gap and the relative gap toward upper bound. The best performance is highlighted in bold.

| Scenarios | Method | Detector | $AP\uparrow$ | $AP_{50}\uparrow$ | $AP_{75}\uparrow$ | AbsGap$\downarrow$ | RelGap$\downarrow$ |
|---|---|---|---|---|---|---|---|
| Upper Bound (Full data) | - | GFL | 40.2 | 58.3 | 43.6 | - | - |
| | | Deformable DETR | 47.0 | 66.1 | 50.9 | | |
| 40 + 40 | LwF (Li & Hoiem, 2017) | GFL | 17.2 | 25.4 | 18.6 | 23.0 | 57.2% |
| | RILOD (Li et al., 2019) | GFL | 29.9 | 45.0 | 32.0 | 10.3 | 25.6% |
| | SID (Peng et al., 2021) | GFL | 34.0 | 51.4 | 36.3 | 6.2 | 15.4% |
| | ERD (Feng et al., 2022) | GFL | 36.9 | 54.5 | 39.6 | 3.3 | 8.2% |
| | PseDet (Ours) | GFL | **38.5** | **54.9** | **41.9** | **1.7** | **4.2%** |
| | CL-DETR (Liu et al., 2023a) | Deformable DETR | 42.0 | 60.1 | 45.9 | 5.0 | 10.6% |
| | SDDGR (Kim et al., 2024) | Deformable DETR | 43.0 | 62.1 | 47.1 | 4.0 | 8.5% |
| | PseDet (Ours) | Deformable DETR | **43.5** | **61.5** | **47.2** | **3.5** | **7.4%** |
| 70 + 10 | LwF (Li & Hoiem, 2017) | GFL | 7.1 | 12.4 | 7.0 | 33.1 | 82.3 |
| | RILOD (Li et al., 2019) | GFL | 24.5 | 37.9 | 25.7 | 15.7 | 39.1 |
| | SID (Peng et al., 2021) | GFL | 32.8 | 49.0 | 35.0 | 7.4 | 18.4 |
| | ERD (Feng et al., 2022) | GFL | 34.9 | 51.9 | 37.4 | 5.3 | 13.2 |
| | PseDet (Ours) | GFL | **39.2** | **55.6** | **42.8** | **1.0** | **2.5%** |
| | CL-DETR (Liu et al., 2023a) | Deformable DETR | 40.4 | 58.0 | 43.9 | 6.6 | 14.0% |
| | SDDGR (Kim et al., 2024) | Deformable DETR | 40.9 | 59.5 | 44.8 | 6.1 | 13.0% |
| | PseDet (Ours) | Deformable DETR | **44.7** | **62.9** | **48.6** | **2.3** | **4.9%** |

For the ground truth, we use QFL as the class loss without any alterations. Integrating the above formulas, we can unify the class loss for both pseudo labels and ground truth as $\mathcal{L}_{cls}$:

$$\mathcal{L}_{cls} = \mathcal{L}_{cls}^{pl} + \mathcal{L}_{cls}^{gt} = -|y - \hat{y}|^{\beta} \left( (1 - y) \log(1 - \hat{y}) + y \log(\hat{y}) \right). \qquad (8)$$

It's worth noting that our method can be utilized with other variations of continual versions of cross-entropy loss. Moreover, it can be easily extended to other domains that leverage pseudo-labels, such as semi-supervised object detection (Wang et al., 2023).

# 4 EXPERIMENTS

## 4.1 EXPERIMENTAL SETTINGS

**Datasets and Evaluation Metric.** MS COCO 2017 (Lin et al., 2014) is an object detection dataset with 80 categories. These categories will be divided into different mutually exclusive sets based on the scenario of the experiment at different steps. The evaluation metrics include standard COCO protocols: $AP, AP_{50}, AP_{75}, AP_S, AP_M$ and $AP_L$; absolute gap (AbsGap) and relative gap (RelGap) between final mAP of incremental learning and mAP of upper-bound; forgetting percentage points (FPP), which is used to evaluate the degree of forgetting for trained categories.

**Experimental Setup.** In our experiments, we apply the method to different scenarios by partitioning the sets of categories into different collections $\mathcal{C}^i$. At each step, we start by training the detector normally using the initial set of categories. In subsequent steps, the model can only access the ground truth of the current step's categories. We mainly focus on the following two scenarios: (a) One-step: $40 + 40, 50 + 30, 60 + 20, 70 + 10$; (b) Multi-step: $40 + 20 \times 2, 40 + 10 \times 4$.

**Implementation Details.** We implemented method on GFL (Li et al., 2020) and Deformable DETR(Zhu et al., 2020) with ResNet-50 image backbone. All experiments are performed on 8 NVIDIA Tesla V100 GPUs, and the basic settings follow the official implementation(Chen et al., 2019). For GFL (Deformable DETR), we set the batch size to 2 (4) per GPU, trained for 12 (50) epochs, and used SGD (AdamW) as the optimizer.

Table 2: **Incremental results ($AP$, %) on COCO benchmark under the multi-step setting**. In the first step, normal training is conducted with 40 categories, followed by the addition of 20 and 10 new categories in the 2-step and 4-step settings each time, respectively.

| Method | Detector | 40+10+10+10+10 | | | | 40+20+20 | |
|---|---|---|---|---|---|---|---|
| | | (40-50) | (50-60) | (60-70) | (70-80) | (40-60) | (60-80) |
| RILOD (Li et al., 2019) | GFL | 25.4 | 11.2 | 10.5 | 8.4 | 27.8 | 15.8 |
| SID (Peng et al., 2021) | GFL | 34.6 | 24.1 | 14.6 | 12.6 | 34.0 | 23.8 |
| ERD (Feng et al., 2022) | GFL | 36.4 | 30.8 | 26.2 | 20.7 | 36.7 | 32.4 |
| PseDet (Ours) | GFL | **39.3** | **37.9** | **37.5** | **37.1** | **38.4** | **38.1** |
| CL-DETR (Liu et al., 2023a) | Deformable DETR | - | - | - | 28.1 | - | 35.3 |
| SDDGR (Kim et al., 2024) | Deformable DETR | 42.3 | 40.6 | 40.0 | 36.8 | **42.5** | 41.1 |
| PseDet (Ours) | Deformable DETR | **42.7** | **41.1** | **41.5** | **41.2** | 42.3 | **42.8** |

Table 3: **Ablation study using the COCO benchmark of 40 classes + 40 classes**. All categories and Old categories represent the performance ($AP/AP_{50}/AP_{75}$, higher is better) evaluated by the model after completing one-step of learning on all 80 categories and the old 40 categories, respectively. The Forgetting Percentage Point (FPP) reflects the performance gap on the initial 40 categories between the start and completion of training, indicating the degree of forgetting with lower values preferred.

| Method | All categories ↑ | | | Old categories ↑ | | | FPP ↓ | | |
|---|---|---|---|---|---|---|---|---|---|
| | $AP$ | $AP_{50}$ | $AP_{75}$ | $AP$ | $AP_{50}$ | $AP_{75}$ | $AP$ | $AP_{50}$ | $AP_{75}$ |
| Fine-tuning | 17.9 | 26.9 | 19.3 | 0.0 | 0.0 | 0.0 | 40.6 | 59.0 | 44.1 |
| + Normal Pseudo Labeling | 22.8 | 33.1 | 24.8 | 26.6 | 37.9 | 29.4 | 14.0 | 21.1 | 14.7 |
| ++ Spatial Enhancement | 29.9 | 42.8 | 32.6 | 31.9 | 44.5 | 35.3 | 8.7 | 14.5 | 8.8 |
| +++ Categorical Adaptive Label Selector | 34.1 | 49.2 | 37.2 | 35.9 | 51.2 | 39.5 | 4.7 | 7.8 | 4.6 |
| ++++ Confidence Score Calibration | **38.5** | **54.8** | **41.9** | **40.8** | **57.5** | **45.1** | **-0.2** | **1.5** | **-1.0** |

## 4.2 Overall Performance

**One-step**. Table 1 shows the incremental performance of PseDet and other approaches on COCO in the one-step scenarios of $40 + 40$ and $70 + 10$. In each scenario, our method demonstrates substantial improvements over state-of-the-art. It is worth mentioning that in the 70+10 scenario, we achieve a 3.8% increase in $mAP$ compared to the previous best method (SDDGR), with an $AbsGap$ between the upper bound narrowing to just 2.3%. Similarly, in $40 + 40$ scenarios, PseDet also maintains leading performance, with improvements of 0.5% in $mAP$, reducing the $AbsGap$ with the upper bound to 3.5%.

**Multi-step**. Table 2 shows the incremental performance of our method and others on COCO in the more challenging multi-step scenarios of $40 + 20 \times 2$ and $40 + 10 \times 4$, which pays more attention to the capability in long-term incremental learning. Unlike other methods that suffer from severe memory decay, our approach exhibits relatively slow forgetting. In the challenging 4-step setting, upon completion of all steps, we achieved a 16.4% improvement in mAP compared to SDDGR. Similarly, in the 3-step setting, we observe an improvement of 1.7%, which validates the effectiveness and the anti-forgetting of our approach.

## 4.3 Ablation Studies

In this section, we conduct ablation experiments using the GFL detector in the 40+40 scenario, and the experimental results are shown in Table 3. Fine-tuning refers to training on new categories without additional operations, resulting in catastrophic forgetting. Normal Pseudo Labeling refers to the utilization of pseudo labels generated by an old detector without any filtering or processing, and mixing them directly with ground truth for training. For Spatial Enhancement, since it is a one-step setting, the experiments improve the quality of pseudo labels solely through spatial enhancement. After implementing the Categorical Adaptive Label Selector and confidence Score Calibration, the model's performance improved by 4.2% and 38.5% respectively in mAP, outperforming previous state-of-the-art methods.

Table 4: The performance of detectors with different queue length of Categorical Adaptive Label Selector under the scenario of 40+40.

| Queue Length | All categories | | | Old categories | | | New categories | | |
|---|---|---|---|---|---|---|---|---|---|
| | $AP$ | $AP_{50}$ | $AP_{75}$ | $AP$ | $AP_{50}$ | $AP_{75}$ | $AP$ | $AP_{.5}$ | $AP_{.75}$ |
| 100 | 38.5 | 54.8 | 41.9 | 40.8 | 57.5 | 45.1 | 36.2 | 52.1 | 38.7 |
| 150 | 38.5 | 54.9 | 41.8 | 40.9 | 58.0 | 45.6 | 36.1 | 51.8 | 38.1 |
| 200 | 38.5 | 54.9 | 41.8 | 40.9 | 57.7 | 45.2 | 36.1 | 52.0 | 38.5 |

Table 5: The performance of detectors with varying $\alpha$ and $\delta$ of the mapping function $sigmoid(\alpha \cdot (s - \delta))$ under the scenario of 40+40.

| Params | | All categories | | | Old categories | | | New categories | | |
|---|---|---|---|---|---|---|---|---|---|---|
| $\alpha$ | $\delta$ | $AP$ | $AP_{50}$ | $AP_{75}$ | $AP$ | $AP_{50}$ | $AP_{75}$ | $AP$ | $AP_{.5}$ | $AP_{.75}$ |
| 10 | 0.6 | 37.1 | 53.1 | 40.3 | 39.2 | 55.9 | 43.2 | 34.9 | 50.3 | 37.4 |
| 10 | 0.5 | **38.5** | **54.8** | **41.9** | **40.8** | **57.5** | **45.1** | **36.2** | **52.1** | **38.7** |
| 10 | 0.4 | 37.9 | 54.2 | 40.9 | 40.3 | 57.2 | 44.4 | 35.4 | 51.3 | 37.5 |
| 5 | 0.5 | 38.3 | 54.5 | 41.5 | 40.7 | 57.3 | 44.9 | 35.9 | 51.7 | 38.1 |
| 20 | 0.5 | 38.1 | 54.6 | 41.6 | 40.6 | 57.4 | 44.7 | 35.9 | 51.8 | 38.4 |

## 4.4 ANALYSIS

**Queue length in Adaptive Label Selector.** Table 4 summarizes model performance across queue lengths. Despite minor fluctuations, overall performance remained steady at 38.5% in $AP$, with slight deviations of only 0.1% in $AP_{50}$ and $AP_{75}$. With a queue length of 100, old category performance decreased slightly while new categories showed a slight improvement, differing by just 0.1% in $AP$. However, longer queues increase computational demands, impacting efficiency. Thus, opting for a smaller value, $N = 100$, which is more practical.

**Coefficients for Score Calibration.** Table 5 presents the performance across various $\alpha$ and $\delta$ settings for the mapping function, with the best performance achieved at $\alpha = 10$ and $\delta = 0.5$. As $\delta$ changes, model performance is affected: increasing $\delta$ reduces attention to pseudo-labels and decreases AP by 1.4%, while decreasing $\delta$ increases $q$, enhancing pseudo-label supervision but introducing noise and causing performance decline. $\alpha$ scales the mapping function, and increasing $\alpha$ widens the quality gap between high and low scores. Compared to $\alpha$, adjusting $\delta$ has a smaller impact on performance.

**Anti-forgetting Performance in Multi-step.** Figure 5 illustrates the model's performance on specific categories under multi-step scenarios. Each line depicts the model's performance on the same subset of categories $\mathcal{C}^i$ at different steps. Starting from accessing ground truth at step $i$, the model exhibits a relatively low forgetting rate in each subset. More detailed analysis can be found in the appendix.

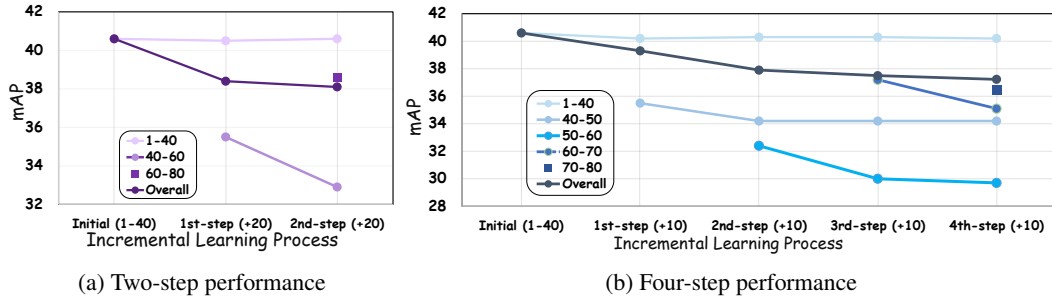

(a) Two-step performance          (b) Four-step performance

Figure 5: **The performance of our model in the multi-step scenario**. Overall indicates an evaluation of the entire set in the current step, while others refer to evaluations of subsets $\mathcal{C}^i$.

## 5 CONCLUSION

In this paper, we revisit the power of pseudo-labeling in continual object detection. Specifically, we identify three critical issues that arise when applying pseudo-labeling. Based on these observations, we propose a simple yet effective framework, namely PseDet. It introduces the spatio-temporal enhancement module to effectively improve the quality of the labels. With the proposed categorical adaptive label selector, PseDet can dynamically decide label filtering thresholds based on the score distributions of different categories. To fully leverage the information from pseudo-labels, we align the confidence scores with the localization quality and integrate them into the supervision. Through extensive experiments conducted on the MS COCO dataset with various settings, we validate the effectiveness of our method, achieving a new state-of-the-art in this era.

## ACKNOWLEDGEMENT

This work was supported by the Anhui Provincial Natural Science Foundation under Grant 2108085UD12. We acknowledge the support of GPU cluster built by MCC Lab of Information Science and Technology Institution, USTC.

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

## A  IMPLEMENTATION DETAILS

**Dataset Preparation and Preprocessing.** There are 80 actual valid categories in COCO 2017, and the category numbers are not consecutive. We employ cocopytool to read annotations and randomly assign category labels to label indices ranging from 0 to 79. In each consecutive experiment, we maintain a fixed random seed and divide them into mutually exclusive sets based on scenarios.

**Training and Evaluation.** During model training, due to the defined category sets, a small portion of images do not contain any annotations. We filter out images that contain neither ground truth nor pseudo-labels. During evaluation, we assess all images in the eval-set but only calculate the metrics for the currently learned categories.

## B  ADDITIONAL EXPERIMENTAL RESULTS

### B.1  NOT FORGETFULNESS, BUT REINFORCEMENT: EMPOWERING MODELS TO MASTER PRIOR KNOWLEDGE

Figure A1 depicts the performance of various methods across different scenarios. Each line represents the performance of the detector on the evaluation set under the one-step scenario. The number of categories in $\mathcal{C}^1$ varies from 40 to 70, while the number of categories in $\mathcal{C}^2$ changes from 40 to 10. The trend of the curves indicates that the previous method exhibits significant forgetting characteristics, whereas our method demonstrates better performance as the amount of knowledge to remember increases. Based on the experimental results above, we have effectively addressed the issue of knowledge forgetting that occurs during the transfer of new knowledge in continual learning, as well as balanced the utilization of model capacity between new and old knowledge.

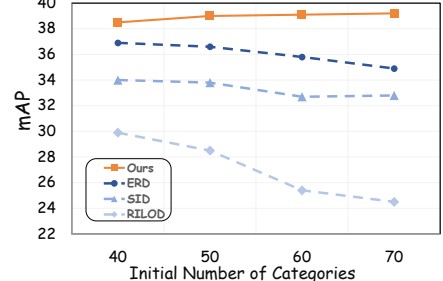

Figure A1: The performance under the one-step scenario. With other methods, the larger the initial set, the faster the forgetting; our method performs better and is suitable for continual learning.

It's quite surprising that despite the expected increase in the absolute value of knowledge forgetting as the number of categories in the first step increases, the performance of the final detector actually improves with our method. We believe this could be attributed to the effect of attention, where we introduced $q$ in the loss calculation, which played a significant role.

Table A1 shows the experimental results of our method based on GFL under the 50+30 and 60+20 scenarios as a supplement to the main paper. All experimental results, including those in Table 3, demonstrate the scalability and robustness of our method, providing further support for our viewpoint: **our method not only alleviates forgetting but also reinforces previous knowledge.**

Table A1: Additional results on COCO benchmark under the one-step setting based on GFL detector. Most experimental results are borrowed from ERD (Feng et al., 2022) paper.

| Detector | Scenarios | Method | $AP\uparrow$ | $AP_{50}\uparrow$ | $AP_{75}\uparrow$ | AbsGap↓ | RelGap↓ |
|---|---|---|---|---|---|---|---|
| | Full data (Upper Bound) | - | 40.2 | 58.3 | 43.6 | - | - |
| | | LwF (Li & Hoiem, 2017) | 5.0 | 9.5 | 4.6 | 35.2 | 87.6% |
| | | RILOD (Li et al., 2019) | 28.5 | 43.2 | 30.2 | 11.7 | 29.1% |
| | 50 classes + 30 classes | SID (Peng et al., 2021) | 33.8 | 51.0 | 36.1 | 6.4 | 15.9% |
| | | ERD (Feng et al., 2022) | 36.6 | 54.0 | 38.9 | 3.6 | 9.0% |
| GFL | | PseDet (Ours) | **39.0** | **55.6** | **42.1** | **1.2** | **3.0%** |
| | | LwF (Li & Hoiem, 2017) | 5.8 | 10.8 | 5.3 | 34.4 | 85.6% |
| | | RILOD (Li et al., 2019) | 25.4 | 38.8 | 26.8 | 14.8 | 36.8% |
| | 60 classes + 20 classes | SID (Peng et al., 2021) | 32.7 | 49.8 | 34.6 | 7.5 | 18.7% |
| | | ERD (Feng et al., 2022) | 35.8 | 52.9 | 38.4 | 4.4 | 11.0% |
| | | PseDet (Ours) | **39.1** | **55.7** | **42.5** | **1.1** | **2.7%** |

## B.2 ROBUST AND HIGH-PERFORMANCE: TACKLING MULTI-STEP CHALLENGES WITH PSEDET

Evaluating the anti-forgetting performance of the model requires not only comparing the performance of the detector on the entire dataset across different steps but, more importantly, tracking the performance of the detector on the same subset. Each line in Figure 5 and Figure A2 tracks the performance of the detector on subset $\mathcal{C}^i$ starting from when the detector learns the ground truth of $\mathcal{C}^i$.

In addition to what we observed in Figure A1, we also noticed a similar phenomenon in multi-step settings, where the more knowledge to be retained, the stronger the anti-forgetting ability. From the Table A4, our method's implementation on Deformable DETR achieved a final performance of 42.8% in AP on the 2-step setting, which is 0.5% higher than the intermediate stage. Similarly, on the 4-step setting, the final performance is 0.1% higher than the intermediate stage.

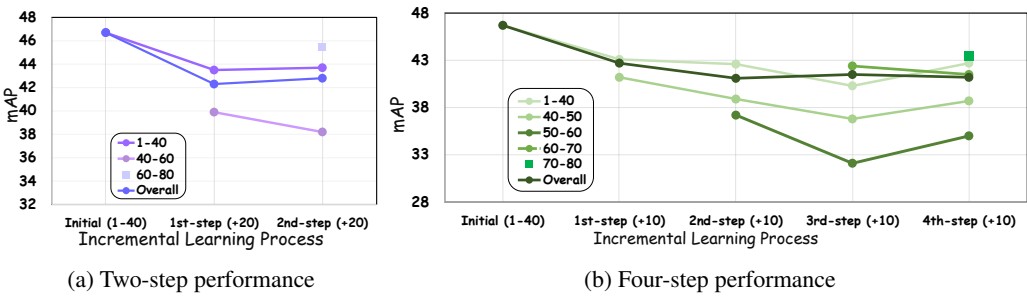

(a) Two-step performance                    (b) Four-step performance

Figure A2: The performance of the Deformable DETR, based on our method, in the multi-step scenario serves as a supplement to Figure 5. Overall indicates an evaluation of the entire set in the current step, while others refer to evaluations of subsets $\mathcal{C}^i$. The stability of each line represents the excellent anti-forgetting ability of our method.

## B.3 LESS IS MORE: NOISE PRESENTS GREATER DISRUPTION TO LEARNING

In object detection, $AP$, as a critical evaluation metric, intuitively, is sometimes used to evaluate the quality of pseudo-labels. As shown in Table A2, when we use filters to remove noise, there is a significant decrease in the metrics. However, after the second stage of learning, the performance of the group of detectors with lower AP noticeably improves.

Although noise contains some useful information, through experimental observation, we believe that appropriately screening out uncertain annotations is helpful for learning and will not result in a loss of knowledge.

Table A2: The evaluation results of the pseudo-labels. In the first step of the 40+40 scenario, the validation set is inferred based on Deformable DETR with or without our pseudo-label selector.

| Setting | $AP$ | $AP_{.5}$ | $AP_{.75}$ | $AP_S$ | $AP_M$ | $AP_L$ |
|---|---|---|---|---|---|---|
| w/o Pseudo-label filtering | 46.0 | 66.6 | 49.4 | 22.9 | 44.7 | 61.6 |
| w/ Pseudo-label filtering | 33.5 | 48.1 | 36.2 | 17.8 | 33.2 | 43.7 |

## C PROOF OF ALGORITHM 1

To start with, we infer pseudo-labels represented by $(c, s, b)$ using old model $\theta_{old}$, where $c$ denotes the class, $s$ indicates the confidence score, and $b$ represents the position coordinates of the bbox.

Based on careful observation, we have noted that, for each class $i$, the model's prediction confidence exhibits a distinct bimodal distribution, comprising both high-confidence regions (positive samples) and low-confidence regions (noise). The data in these regions respectively display a "Gaussian bell" shape centered around specific means $\mu$:

High-confidence peak: This peak indicates that the model is highly confident in its detection results, representing high-quality pseudo-labels. These samples are characterized by clear features and high accuracy, clustering around $\mu_T$.

Low-confidence peak: Typically, this peak reflects the model's uncertain predictions or samples that are difficult to distinguish. In this region of distributions, the pseudo-labels are mostly inaccurate noise, clustering around $\mu_F$.

The confidence score follows a bimodal distribution, and its probability density can be expressed as:

$$p(s) = \pi_T \mathcal{N}(s \mid \mu_T, \sigma_T^2) + \pi_F \mathcal{N}(s \mid \mu_F, \sigma_F^2) \tag{A1}$$

Let $S_i \in \mathbb{R}^n$ represents the set of confidence scores for the $i-th$ class. In E-step of the Expectation-Maximization (EM) algorithm, we assign each data point to the cluster in which it has the highest posterior. The posterior is:

$$p(c_j = k \mid s_j) = \frac{\pi_k \mathcal{N}(s_j \mid \mu_T, \sigma_T^2)}{\pi_T \mathcal{N}(s_j \mid \mu_T, \sigma_T^2) + \pi_F \mathcal{N}(s_j \mid \mu_F, \sigma_F^2)} \tag{A2}$$

For the convenience of analysis, we can obtain the reciprocal of the posterior:

$$\begin{aligned} p(c_j = True \mid s_j)^{-1} &= 1 + \frac{\pi_F \mathcal{N}(s_j \mid \mu_F, \sigma_F^2)}{\pi_T \mathcal{N}(s_j \mid \mu_T, \sigma_T^2)} \\ &= 1 + \frac{\pi_F}{\pi_T} \exp(\frac{1}{2\sigma_T^2}(s_i - \mu_T)^2 - \frac{1}{2\sigma_F^2}(s_i - \mu_F)^2) \end{aligned} \tag{A3}$$

Here, K-means aims to partition data into two clusters by minimizing the Within-Cluster Sum of Squares:

$$J = \sum_{i \in C_k} (\|s_i - \mu_T\|^2 + \|s_i - \mu_F\|^2) \tag{A4}$$

In this distribution, samples can be considered to have soft assignments. However, we can use K-means which is a type of hard assignment to assign sample $i$ to the cluster $k$ ($k \in \{True, False\}$):

$$c_k = \arg \min_k \|s_j - \mu_k\|^2 \tag{A5}$$

By combining Equations A3 and A5, we obtain:

$$\|s_i - \mu_k\|^2 \propto p^{-1} \tag{A6}$$

Maximizing the posterior probability $p(c_j = k \mid s_j)$ is equivalent to minimizing the $\|s_j - \mu_k\|^2$.

K-Means can be considered a special case of the Expectation-Maximization (EM) algorithm with hard assignments, aimed at modeling the bimodal distribution of scores for true and false clusters. In addition, K-means clustering is often highly sensitive to the choice of initial centroids. Therefore, we repeat the clustering process multiple times.

## D  ADDITIONAL ANALYSIS ON MEMORY AND TIME OVERHEAD.

(1) GPU Memory. No additional GPU Memory is needed. The current methods, such as ERD (Feng et al., 2022), require loading two models during each training iteration: one is the original model used to infer the logistic of a sample, and the other is the model from the current new stage. Our method generates pseudo labels in an offline manner. This means that during training, our method is the same as the regular training and does not require any additional GPU memory. For example, based on GFL, our method occupies approximately 4.5 GB of GPU memory per GPU, whereas ERD requires more than 10 GB per GPU, nearly double the GPU memory compared to our method. (2) Time Efficiency. Additional time cost is negligible. Diffusion-based methods (Kim et al., 2024) require additional time to train a diffusion model and generate new samples, while distillation-based methods (Feng et al., 2022) necessitate an extra inference step in each training iteration. In contrast, our approach infers pseudo labels once in an offline manner. This pseudo-label set can then be repeatedly used in

subsequent training sessions. If training is conducted only once, the additional time overhead for this offline inference is approximately limited to 5%. As the number of training steps increases, the time efficiency advantage of our method becomes even more significant. (3) Disk Usage: The additional disk usage required by our method is negligible. After inferring pseudo labels with the previous weights, the labels can be stored. It takes only a few bytes per sample. For the COCO dataset, the additional pseudo label .pkl files occupy approximately 10 MB of disk space.

Table A3: Comparison about Memory and Time.

| Method | Core | GPU Memory | Time Efficiency | Disk Usage |
|---|---|---|---|---|
| ERD | KD-Based | Two Models loaded | An additional inference for one training iteration | Last step checkpoint. |
| SDDGR | Diffusion-based | Normal Train (One Model) | Additional time to generate new samples with diffusion model. | Additional samples generated by diffusion model. |
| PseDet (Ours) | Pseudo-Label | Normal Train (One Model) | Additional time to inference Pseudo-Labels. | Every step checkpoint but only a few hundred megabytes are required for each. |

## E  IMPLEMENTATION IN SEGMENTATION TASK

We implemented our method based on SegFormer (Xie et al., 2021) and the basic settings follow the official implementation. We selected the ADE20K (Zhou et al., 2017) as our experimental dataset. This dataset presents a significant challenge due to its extensive category count of 150. It offers significant reference value for the implementation of class incremental learning.

In our experiment, we design two scenarios: 100+50 as the one-step setting and 100+25+25 as the multi-step setting. The evaluation metrics include mIoU (higher is better); absolute gap (AbsGap, lower is better) and relative gap (RelGap, lower is better) between final mIoU of incremental learning and mIoU of upper-bound; forgetting percentage points (FPP, lower is better), which is used to evaluate the degree of forgetting for trained categories.

Table A4: Incremental results on ADE20K benchmark based on SegFormer.

| Method | 100+150 Scenario | | | | | | 100+25+25 Scenario | | | | | | |
|---|---|---|---|---|---|---|---|---|---|---|---|---|---|
| | Overall | 1-100 | 101-150 | AbsGap↓ | RelGap↓ | FPP↓ | Overall | 1-100 | 101-125 | 126-150 | AbsGap↓ | RelGap↓ | FPP↓ |
| Upper Bound | 37.41 | 41.23 | 29.77 | - | - | - | 37.41 | 41.23 | 32.94 | 26.60 | - | - | - |
| Catastrophic Forgetting | 0.88 | 0 | 2.63 | 36.53 | 97.64 | 43.11 | 0.23 | 0 | 0 | 1.36 | 37.18 | 99.38 | 43.11 |
| PseSeg (Ours) | 36.18 | 41.43 | 25.69 | 1.23 | 3.29 | 1.68 | 35.00 | 40.63 | 25.44 | 22.04 | 2.41 | 6.44 | 0.60 |

Considering that segmentation tasks involve dense predictions and that, for SegFormer, there are no classification and regression branches as seen in detection tasks, it is unnecessary to consider decoupling classification and regression tasks within the context of continual learning. Consequently, pseudo-label-based methods may yield better performance when applied to segmentation tasks. Consistent with the experimental results, the model trained through continual learning exhibits an absolute gap of only 1.23 mIoU in the one-step scenario and 2.41 mIoU in the multi-step scenario between our method and the upper bound. FPP gets 1.68 mIoU and 0.60 mIoU in the one-step and multi-step settings respectively, reflecting the excellent resistance to forgetting of our method.

## F  ADDITIONAL VISUALIZATION RESULTS

Figure A3 displays the visualization of pseudo-labels at the final stage in the 70+10 scenario. It can be observed that the detector still maintains a good detection performance for objections with challenging difficulty levels. The advantage of the pseudo-label strategy also lies in the detector's ability to discover and annotate objects that were not manually labeled. As can be seen from the last two images in Figure A3, flowers that are occluded or smaller horses that were not annotated were still detected by the detector. This, to some extent, mitigates the loss in memory.

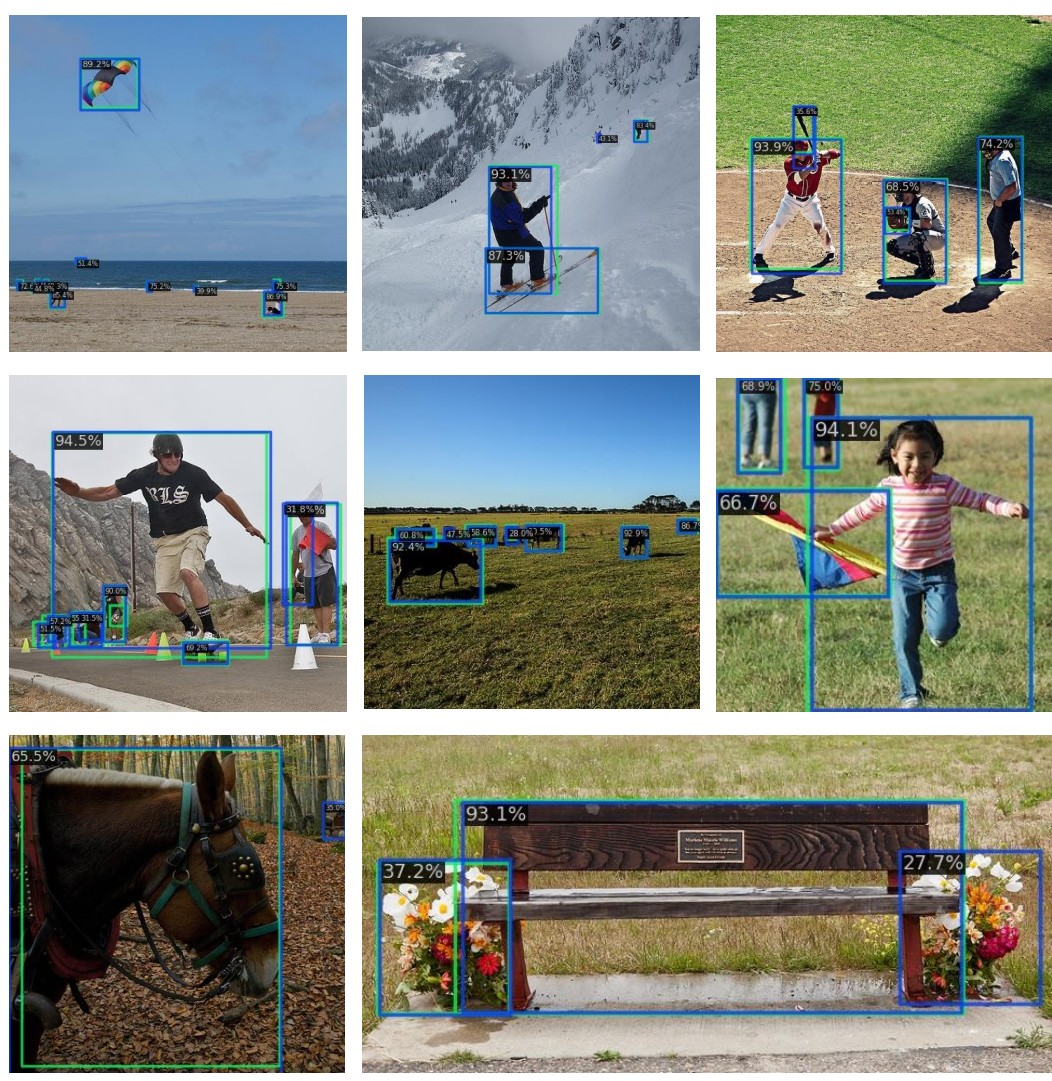

Figure A3: **The visualization results of pseudo-labels.** The ground truth is depicted with green bounding boxes, while pseudo labels are indicated by blue bounding boxes.

