# OpenReview forum: "PseDet: Revisiting the Power of Pseudo Label in Incremental Object Detection"
_ICLR.cc/2025/Conference — ICLR 2025 Poster_

### Official Review · Reviewer_qjDZ · 2024-11-02

**Soundness:** 3
**Presentation:** 2
**Contribution:** 2
**Rating:** 6
**Confidence:** 5

**Summary:**

Incremental Object Detection (IOD) expands object detectors without forgetting prior knowledge. Existing methods rely on knowledge distillation but underutilize the teacher model's insights. This paper identifies three problems in pseudo-labeling: limited pseudo label quality, fixed label filtering thresholds, and weak alignment of confidence scores with localization quality. To address these problems, this paper proposes PseDet, which includes a spatiotemporal enhancement module to handle noisy data, a Categorical Adaptive Label Selector for dynamic class-specific thresholds, and confidence score calibration improves label confidence scores alignment with localization.

**Strengths:**

1. The proposed modules look useful for incremental object detection, including spatio-temporal enhancement module, categorical adaptive label selector, confidence score calibration supervision.
2. The motivation of categorical adaptive label selector is clear.
3. The experimental performance of the proposed PseDet looks good compared to the SOTAs.
4. According to the ablation study in Table 3, the proposed modules are effective.

**Weaknesses:**

1. It is unclear why the proposed spatio-temporal enhancement module is effective. With the empirical good results, please provide more inside explanation.
2. The algorithm of categorical adaptive label selector is provided. Please provide some caption and formula to further describe it for clearer presentation.
3. In categorical adaptive label selector, why k-means of the confidence scores can be a good threshold?
4. The difference between the proposed confidence score calibration supervision and the IoU-aware classification score in VFNet [A].

Overall, the proposed modules are effective, but lack inside explanations. I will give a rating of 5. If more inside explanations are provided, I will give a higher rating.

[A] VarifocalNet: An IoU-aware Dense Object Detector.

**Questions:**

see the weaknesses

---

> ### Author Response · Authors · 2024-11-20
> **Responses to Official Review by Reviewer qjDZ: Part[1/3]**
>
> Dear Reviewer qjDZ,
>
> Thanks for the valuable feedback. We have tried our best to address all concerns in the last few days. Please see our responses below one by one:
>
> > **W1: About inside explanation of the spatio-temporal enhancement module.**
>
> The spatio-temporal enhancement module aims to strengthen continual learning by addressing both the temporal and spatial domains. Our efforts are primarily overcoming two key challenges: improving the quality of pseudo-labels and enhancing the ability to effectively learn from these pseudo-labels.
>
> **(1)** Temporal Enhancement: As continuous learning progresses, the model suffers from forgetting of knowledge related to old categories. In Figure 3 of the manuscript, forgetting occurs between the initial step and the second step across all categories. **The model's best performance on a category is observed in the initial step of learning this category.** Therefore, for pseudo-labels of a specific category, we select the one closest to the initial step on the timeline, as the quality of the pseudo-labels is highest at this point. Temporal Enhancement is used to strengthen a model's ability to resist forgetting in multi-step scenarios. As shown in Figures 5(a) and (b), with temporal enhancement, the FPP on the initial categories decreases to 0.3 $AP$ and 0.5 $AP$ in the two-step and four-step settings, respectively.
>
> **(2)** Spatio Enhancement: Spatial augmentation transformation includes horizontal flip and scaling.
>
> + **Models often exhibit a certain spatial bias, where features in specific positions are more easily captured by the model by flipped images.** Horizontal flipping can mitigate this bias effect. When we evaluate the GFL model, the performance is 41.5 AP under vanilla settings. After applying horizontal flipping, the performance is 30.8 AP. However, if we use NMS to fuse these two sets, the result improves to 42.2 AP.
>
> + **Scaling allows the model to have a more flexible perspective, as the size of objects in an image is crucial for model predictions. Scaling down the image helps in recognizing objects that occupy a larger area in the image.** Experimental result shows the performance improves to 46.8 $AP_l$ after scale down. Scaling up the image help in identifying smaller objects within the image. The performance improves to 18.6 $AP_m$ after scale up. After integrating all the transformations, the performance improved to 43.1 $AP$.The table below shows the performance after various spatial transformations, where $AP_s$, $AP_m$, $AP_l$ respectively represent the AP for objects of small, middle, and large sizes. The best result is marked with **bold**.
>
> |  Method | $AP$ | $AP_{50}$ | $AP_{75}$ | $AP_{s}$ | $AP_{m}$ | $AP_{l}$ |
> | --- | --- | --- | --- | --- | --- | --- |
> | vanilla | 41.5 | 66.2 | 43.8 | 16.5 | 29.3 | 45.6 |
> | scale down | 40.1 | 63.6 | 42.6 | 8.3 | 24.3 | 46.8 |
> | scale up | 33.1 | 58.1 | 32.8 | 18.6 | 28.8 | 34.3 |
> | Spatio Enhancement  | **43.1** | **66.8** | **44.1** | **19.2** | **29.7** | **47.3** |

---

> ### Author Response · Authors · 2024-11-20
> **Responses to Official Review by Reviewer qjDZ: Part[2/3]**
>
> > **W2: Give more detailed explanation to algorithm of categorical adaptive label selector.**
>
> We will add more details of algorithm 1 in the revision. Further explanations and proofs regarding Algorithm 1 are also provided in the response of W3. Thanks for this suggestion!
>
> > **W3: The reason why k-means of the confidence scores can serve as a good threshold in label selector.**
>
> To start with, we infer pseudo-labels represented by $(c, s, b)$ using old model $\theta_{old}$, where $c$ denotes the class, $s$ indicates the confidence score, and $b$ represents the position coordinates of the bounding box.
> Based on careful observation, we have noted that, for each class $i$, the model's prediction confidence exhibits a distinct bimodal distribution, comprising both high-confidence regions (positive samples) and low-confidence regions (noise). The data in these regions respectively display a "Gaussian bell" shape centered around specific means $\mu$:
> + High-confidence peak: **This peak indicates that the model is highly confident in its detection results, representing high-quality pseudo-labels.** These samples are characterized by clear features and high accuracy, clustering around $\mu_T$.
> + Low-confidence peak: Typically, **this peak reflects the model's uncertain predictions or samples that are difficult to distinguish**. In this region of distributions, the pseudo-labels are mostly inaccurate noise, clustering around $\mu_F$.
>
> The confidence score follows a bimodal distribution, and its probability density can be expressed as:
>
> $p(s) = \pi_T \mathcal{N}(s \mid \mu_T, \sigma_T^2) + \pi_F \mathcal{N}(s \mid \mu_F, \sigma_F^2)\quad[1]$
>
> Let $S_i \in \mathbb{R}^n$ represents the set of confidence scores for the $i-th$ class. In E-step of the Expectation-Maximization (EM) algorithm, we assign each data point to the cluster in which it has the highest posterior. The posterior is:
>
> $p(c_j=k \mid s_j) = \frac {\pi_k \mathcal{N}(s_j \mid \mu_T, \sigma_T^2) } {\pi_T \mathcal{N}(s_j \mid \mu_T, \sigma_T^2) + \pi_F \mathcal{N}(s_j \mid \mu_F, \sigma_F^2)}\quad[2]$
>
> For the convenience of analysis, we can obtain the reciprocal of the posterior:
>
> $p(c_j= True \mid s_j)^{-1} == 1 + \frac{\pi_F \mathcal{N}(s_j \mid \mu_F, \sigma_F^2)}{\pi_T\mathcal{N}(s_j \mid \mu_T, \sigma_T^2) }
> ==1 + \frac{\pi_F}{\pi_T} \exp({ \frac{1}{2\sigma_T^2}(s_i-\mu_T)^2 - \frac{1}{2\sigma_F^2}(s_i-\mu_F)^2})\quad[3]$
>
> Here, K-means aims to partition data into two clusters by minimizing the Within-Cluster Sum of Squares:
>
> $J = \sum_{i \in C_k} (\Vert{s_i - \mu_T}\Vert^2+\Vert{s_i - \mu_F}\Vert^2)
> \quad[4]$
>
> In this distribution, samples can be considered as having soft assignments. However, we can use K-means which is a type of hard assignment to assign sample $i$ to the cluster$k(k \in{\{True,False}\})$:
>
> $c_k = \arg\min_k \| s_j - \mu_k \|^2\quad[5]$
>
> By combining Equations (3) and (5), we obtain:
>
> $\| s_i - \mu_k \|^2 \propto p^{-1}\quad[6]$
>
> Maximizing the posterior probability $p(c_j=k \mid s_j)$ is equivalent to minimizing the $\| s_j - \mu_k \|^2$.
>
> **K-Means can be considered a special case of the Expectation-Maximization (EM) algorithm with hard assignments, aimed at modeling the bimodal distribution of scores for true and false clusters.**
>
> Finally, We distinguish between true positive clusters $C_T$ and false positive clusters $C_F$based on the criterion $\mu_T > \mu_F$.
> In addition, K-means clustering is often highly sensitive to the choice of initial centroids. Therefore, we repeat the clustering process multiple times.

---

> ### Author Response · Authors · 2024-11-20
> **Responses to Official Review by Reviewer qjDZ: Part[3/3]**
>
> > **W4: The difference between the Confidence Score Calibration Supervision and the IoU-aware Classification Score in VFNet.**
>
> The differences will be explained in the following two parts:
>
> **(1) Differences in Usage**:
> - In VFNet, IACS (IoU-aware Classification Score) reflects the comprehensive scores of the samples predicted by the trained model during the training, replacing the role of confidence scores used in other models. In Varifocal Loss, IACS replaces the prediction score of the model in focal loss. IACS is also used as a basis for ranking and selecting samples in modules such as assigner.
> - In our method, the scores of the pseudo labels reflect their quality, and the pseudo labels are used for supervision. We use the calibrated score $q$ to represent the quality of the GT labels, with the IoU $\tau$ (between the generated bbox and its GT bbox) jointly indicating labels $\hat{y} = \tau \cdot q$.
>
> **In our method, calibrated score is used to generate continuous GT labels; in VFLNet, IACS is a comprehensive score used for assigners, ranking, and calculating loss, etc.**
>
> **(2) Differences in Essence**:
> - In VFNet, IACS is generated by merging object predict confidence and localization accuracy into a single composite score. The score is based on the prediction of a sample's classification and bbox. It is obtained through learning
> - In our method, the score is derived from a mathematical mapping of the model's confidence score, with the aim of making the score more relevant to the quality of pseudo-labels.
>
> In summary, **Confidence Score Calibration Supervision is specifically designed for pseudo-label based tasks, making it more suitable for pseudo-labeling methods. It can be effectively applied to other tasks and models, demonstrating superior performance in continual learning tasks.** However, IACS is specifically tailored for object detectors with classification and regression branches. It varies across different models and tasks, making it unsuitable for pseudo-labeling tasks. Additionally, it is difficult to extend it to other pseudo-label based tasks such as segmentation or classification.

---

### Official Review · Reviewer_T1Rk · 2024-11-04

**Soundness:** 2
**Presentation:** 3
**Contribution:** 3
**Rating:** 6
**Confidence:** 3

**Summary:**

The paper addresses the challenge of Incremental Object Detection (IOD), where object detectors need to learn new classes without forgetting previously learned ones. The authors propose PseDet, a framework that improves pseudo-labeling in three key ways:

1. Spatio-Temporal Enhancement Module: Improves pseudo-label quality by combining predictions from different spatial transformations and temporal steps.
2. Categorical Adaptive Label Selector: Automatically determines class-specific thresholds for filtering pseudo-labels using K-means clustering, replacing fixed thresholds.
3. Confidence Score Calibration: Aligns confidence scores with localization quality through non-linear mapping for better supervision.

The approach achieves state-of-the-art performance on COCO dataset, with 43.5+/41.2+ mAP under 1/4-step incremental settings.

**Strengths:**

1. The paper identifies and addresses three critical issues in current pseudo-labeling approaches
2. The proposed solutions are practical and well-justified
3. The framework is modular and can potentially be integrated with other detection systems
4. PseDet achieves large performance improvements on several benchmarks.

**Weaknesses:**

1. Algorithm 1 is not clear. What does D^c_T and D^c_F mean? Why the output of K-means is D^c_T and D^c_F? I suggest the authors to clarify these.
2. The non-linear mapping function for confidence calibration seems empirically determined without strong theoretical justification. Can PseDet generalize well to other datasets?

**Questions:**

See weakness above

---

> ### Author Response · Authors · 2024-11-20
> **Responses to Official Review by Reviewer T1Rk: Part[1/3]**
>
> Dear Reviewer T1Rk,
>
> Thank you for the valuable feedback. Over the past few days, we have worked diligently to address all of your concerns. Please see our detailed responses to each point below:
>
> > **W1: Algorithm 1 is not clear.**
>
> **(1)** $D^c_T$ represents **the set of True positive samples** for category $c$ and serves as the pseudo-label set to be used for training. $D^c_F$represents **the set of False Positive samples** for category $c$, which are regarded as noise and will be discarded. $D^c_T$ and $D^c_F$ are obtained through clustering using k-means on the set of confidence scores for category $c$.
> **(2)** Theoretical proof: To start with, we infer pseudo-labels represented by $(c, s, b)$ using old model $\theta_{old}$, where $c$ denotes the class, $s$ indicates the confidence score, and $b$ represents the position coordinates of the bounding box.
> Based on careful observation, we have noted that, for each class $i$, the model's prediction confidence exhibits a distinct bimodal distribution, comprising both high-confidence regions (positive samples) and low-confidence regions (noise). The data in these regions respectively display a "Gaussian bell" shape centered around specific means $\mu$:
> + High-confidence peak: **This peak indicates that the model is highly confident in its detection results, representing high-quality pseudo-labels.** These samples are characterized by clear features and high accuracy, clustering around $\mu_T$.
> + Low-confidence peak: Typically, **this peak reflects the model's uncertain predictions or samples that are difficult to distinguish**. In this region of distributions, the pseudo-labels are mostly inaccurate noise, clustering around $\mu_F$.
>
> The confidence score follows a bimodal distribution, and its probability density can be expressed as:
>
> $p(s) = \pi_T \mathcal{N}(s \mid \mu_T, \sigma_T^2) + \pi_F \mathcal{N}(s \mid \mu_F, \sigma_F^2)\quad[1]$
>
> Let $S_i \in \mathbb{R}^n$ represents the set of confidence scores for the $i-th$ class. In E-step of the Expectation-Maximization (EM) algorithm, we assign each data point to the cluster in which it has the highest posterior. The posterior is:
>
> $p(c_j=k \mid s_j) = \frac {\pi_k \mathcal{N}(s_j \mid \mu_T, \sigma_T^2) } {\pi_T \mathcal{N}(s_j \mid \mu_T, \sigma_T^2) + \pi_F \mathcal{N}(s_j \mid \mu_F, \sigma_F^2)}\quad[2]$
>
> For the convenience of analysis, we can obtain the reciprocal of the posterior:
>
> $p(c_j= True \mid s_j)^{-1} == 1 + \frac{\pi_F \mathcal{N}(s_j \mid \mu_F, \sigma_F^2)}{\pi_T\mathcal{N}(s_j \mid \mu_T, \sigma_T^2) }
> ==1 + \frac{\pi_F}{\pi_T} \exp({ \frac{1}{2\sigma_T^2}(s_i-\mu_T)^2 - \frac{1}{2\sigma_F^2}(s_i-\mu_F)^2})\quad[3]$
>
> Here, K-means aims to partition data into two clusters by minimizing the Within-Cluster Sum of Squares:
>
> $J = \sum_{i \in C_k} (\Vert{s_i - \mu_T}\Vert^2+\Vert{s_i - \mu_F}\Vert^2)
> \quad[4]$
>
> In this distribution, samples can be considered as having soft assignments. However, we can use K-means which is a type of hard assignment to assign sample $i$ to the cluster$k(k \in{\{True,False}\})$:
>
> $c_k = \arg\min_k \| s_j - \mu_k \|^2\quad[5]$
>
> By combining Equations (3) and (5), we obtain:
>
> $\| s_i - \mu_k \|^2 \propto p^{-1}\quad[6]$
>
> Maximizing the posterior probability $p(c_j=k \mid s_j)$ is equivalent to minimizing the $\| s_j - \mu_k \|^2$.
>
> **K-Means can be considered a special case of the Expectation-Maximization (EM) algorithm with hard assignments, aimed at modeling the bimodal distribution of scores for true and false clusters.**
>
> Finally, We distinguish between true positive clusters $C_T$ and false positive clusters $C_F$based on the criterion $\mu_T > \mu_F$.
> In addition, K-means clustering is often highly sensitive to the choice of initial centroids. Therefore, we repeat the clustering process multiple times.

---

> ### Author Response · Authors · 2024-11-20
> **Responses to Official Review by Reviewer T1Rk: Part[2/3]**
>
> > **W2.1: About the non-linear mapping function for confidence calibration.**
>
> Below, we offer some deeper insights with the hope of addressing your question. The non-linear mapping function is the solution we proposed based on our observations.
>
> The model's predicted confidence scores lie within the set [0,1]. However, it's not always true that the higher the confidence score of a pseudo-label indicates a better quality. Vice versa. If we use the IoU between the pseudo-label and the GT to represent the quality of the pseudo-label, we can observe that **the confidence scores and quality do not exhibit a strong linear relationship which is shown in Figure 4(a).**
>
> Typically, a pseudo-label with a score of 0.7 is very close to the ground truth, while one with a score of 0.3 differs significantly from the ground truth. We prefer high-quality labels are adjusted to 0.9 or higher, and low-quality labels or noise to 0.1 or lower in soft label supervision. **This non-linear mapping aims to reduce noise interference and enhance the learning of valuable information.** The reason we don't just remove low-scoring labels right away is that, after using the Adaptive Label Selector, we can still get some helpful information from them.
>
> For this purpose, we carried out many experiments using different non-linear functions. **We selected sigmoid, tanh, and logarithmic functions, and we performed mathematical transformations on these functions to better suit the calibration of scores.**
>
> | Method | Overall Performance |
> | --- | --- |
> | w\o calibration | 34.1 49.2 37.2 |
> | $tanh(\alpha \cdot (s - \delta) )$ | 38.1 55.1 41.7  |
> | $\ln (\alpha \cdot (s - \delta) ) + \beta$ | 35.7 50.4 37.9 |
> | $sigmoid(\alpha \cdot (s - \delta) )$ | 38.5 54.9 41.9 |
>
> **As indicated by the experimental results, we require a function that is convex in high-score regions and concave in low-score regions, which is why we have chosen the sigmoid function.** As we expected, if a function can boost the scores of high-quality samples and lower those of low-quality ones, increasing the gap between them under supervision, it can be effective. However, tuning the hyperparameters of the selected functions to attain optimal performance requires adjustments, similar to adjusting the learning rate across different datasets. **Nevertheless, in our extended experiments, we employed the same hyperparameters on VOC and COCO datasets and achieved good performance as well.**

---

> ### Author Response · Authors · 2024-11-20
> **Responses to Official Review by Reviewer T1Rk: Part[3/3]**
>
> > **W2.2: Implementation the method on other datasets.**
>
> Thank you for your highly valuable questions, which have been instrumental in enhancing our work. Over the past few days, we have implemented our method on the **VOC 2007**, designing various settings including **one-step: 10+10, 15+5, and 19+1**; as well as **multi-step: 10+5+5 and 5+5+5+5**. The basic settings and pipeline are the same as those presented in our manuscript, and we choose GFL[ref1] as the detector. We replicated ERD[ref2] using the official release repository. Experimental results demonstrate that our method continues to exhibit strong performance and robustness. The source code is released in [anonymous code repository](https://anonymous.4open.science/r/PseDet-F1CB).
>
> The table below shows Incremental results ($mAP$, %) under the one-step setting. AbsGap (lower is better) and RelGap (lower is better) represents the absolute gap and the relative gap toward upper bound.
>
> | Scenarios | Method | $AP\uparrow$ | $AP_{50}\uparrow$ | $AP_{75}\uparrow$ | $AbsGap\downarrow$ | $RelGap\downarrow$ |
> | :---: | :---: | :---: | :---: | :---: | :---: | :---: |
> | Upper Bound | - | 41.5 | 66.2 | 43.8 | - | - |
> | 10 + 10 | ERD | 34.2 | 60.1 | 37.2 | 7.3 | 17.6% |
> | | PseDet | **37.4** | **61.2** | **39.1** | **4.1** | **9.8%** |
> | 15 + 5 | ERD | 36.4 | 58.6 | 38.7 | 5.1 | 12.3 |
> | | PseDet | **39.8** | **64.7** | **41.3** | **1.7** | **4.1%** |
> | 19 + 1 | ERD | 37.3 | 59.2 | 38.3 | 4.2 | 10.1% |
> | | PseDet | **40.0** | **65.0** | **41.1** | **1.5** | **3.6%** |
>
>
> The table below shows the results ($mAP$, %) under the scenarios of 10+5+5.
>
> | Method | 10-15 | 15-20 |
> | --- | :---: | :---: |
> | ERD | 34.7 | 28.6 |
> | PseDet | **36.2** | **34.2** |
>
> The table below shows the results ($mAP$, %) under the scenarios of 5+5+5+5.
> | Method | 5-10 | 10-15 | 15-20 |
> | --- | :---: | :---: | :---: |
> | ERD | 30.1 | 27.9 | 23.8 |
> | PseDet | **32.6** | **31.5** | **30.2** |
>
> Notably, in the multi-step setting, we observe that ERD exhibits rapid forgetting after the second step, but our method forgets very slowly. This phenomenon is also related to the anti-noise methods we have designed during the knowledge transfer process.
>
> > [ref1]Generalized Focal Loss: Learning Qualified and Distributed Bounding Boxes for Dense Object Detection. NeurIPS 2020
> > [ref2] Overcoming Catastrophic Forgetting in Incremental Object Detection via Elastic Response Distillation. CVPR 2022

---

### Official Review · Reviewer_aL8R · 2024-11-06

**Soundness:** 2
**Presentation:** 3
**Contribution:** 2
**Rating:** 5
**Confidence:** 4

**Summary:**

This paper presents PseDet, a framework for Incremental Object Detection (IOD) designed to address limitations in existing methods, such as low-quality pseudo-labels, fixed filtering thresholds, and misaligned confidence scores. PseDet introduces a spatio-temporal enhancement module to handle noise from previous models, a Categorical Adaptive Label Selector to adjust thresholds by class, and a non-linear score mapping for better localization accuracy. Experiments on COCO benchmarks show that PseDet achieves state-of-the-art performance.

**Strengths:**

1. This paper is well-structured and easy to follow, with a clear motivation. The three main components—Spatio-Temporal Enhancement, Categorical Adaptive Label Selector, and Confidence Score Calibration Supervision—are thoughtfully designed to address specific limitations in existing methods.

2. The observations presented, such as the effect of varying hard thresholds (Figure 2) and the IoU-score distribution for pseudo-labels before and after mapping (Figure 4), are straightforward yet insightful. These findings offer valuable perspectives that could inspire future research directions.

**Weaknesses:**

1.The proposed three components each address distinct issues, but the relationships between them are not well explored in the paper. A more thorough discussion of their interconnections would strengthen the overall framework.

2.In the Spatio-Temporal Enhancement module, typically only the t−1 model can be stored in memory. Feeding varied types and sizes into all previous models would incur substantial memory costs and may create an unfair comparison with other methods. Including an analysis of training time and memory costs would be valuable for a more comprehensive evaluation.

**Questions:**

1.Equation 2 is unclear in its practical application. It’s not evident how the augmentation transformation set A for the image x can be reversed for the label y to obtain pseudo-labels after fusion. A more detailed explanation of this process would be helpful.

2.It would be interesting to see a discussion on applying the proposed methods to other tasks, such as image classification and semantic segmentation, where pseudo-labels are also used.

---

> ### Author Response · Authors · 2024-11-20
> **Responses to Official Review by Reviewer aL8R: Part[1/4]**
>
> Dear Reviewer aL8R,
>
> Thank you for the valuable feedback. Over the past few days, we have worked hard to address all of your concerns. Please see our detailed responses to each point below:
>
> > **W1: About the relationship between three components.**
>
> **Their contributions to enhancing the model's capabilities are orthogonal, but their ultimate goal is the same**: to reduce the noise associated with using pseudo labels as knowledge carriers in continual learning and to enhance the model's ability to learn useful knowledge from them. **We explored the interconnection between them in Table 3 of the manuscript, and in the following table, we conducted a more detailed ablation study using the same settings as in Table 3.**
>
> | Fine-Tuning | Normal Pseudo Labeling | Spatial Enhancement | Categorical Adaptive Label Selector | Confidence Score Calibration | All Categories $\uparrow$ | Old Categories$\uparrow$ | FPP $\downarrow$ |
> | :---:  | :---:  | :---:  | :---:  | :---:  | :---: | :---: | :----: |
> | $\checkmark$ |  | | | | 17.9 | 0.0 | 40.6 |
> |  | $\checkmark$ |  | | | 22.8 | 26.6 | 14.0 |
> |  | | $\checkmark$ |  | | 29.9 | 31.9 | 8.7 |
> |  | |  | $\checkmark$ | | 25.7 | 29.9 | 10.7 |
> |  | |  |  | $\checkmark$ | 28.8 | 32.0 | 18.6 |
> |  | | $\checkmark$ |  | $\checkmark$ | 31.2 | 32.3 | 8.3 |
> |  | |  | $\checkmark$ | $\checkmark$ | 37.4 | 39.2 | 1.4 |
> |  | | $\checkmark$ | $\checkmark$ |  | 34.1 | 35.9 | 4.7 |
> |   | | $\checkmark$ | $\checkmark$ | $\checkmark$ | $38.5$ | $40.8$ | $-0.2$ |
>
> From the ablation experiments in the table, **all three modules individually enhance the model's performance**. The Spatial Enhancement significantly improves the performance on Overall and FPP. However, the Categorical Adaptive Label Selector plays a more significant role in reducing forgetting of old categories, and Confidence Score Calibration improves the overall performance of new steps, which is related to score-guided supervision.
>
> These three modules collectively constitute a comprehensive methodology, yet each module optimizes different aspects of pseudo labels. The following is illustrated from three distinct perspectives:
>
> The STE module **enhances the accuracy of individual pseudo-labels**. By employing spatial enhancement, it improves the bbox precision of pseudo-labels, and through dynamic sampling in the temporal domain, it selects the highest quality pseudo-labels across the step. The table below illustrates the effectiveness of the STE module. We evaluate the $AP$,$AP_{50}$, $AP_{75}$ on the training dataset after the first training phase to show the quality improvement of the pseudo labels with and without STE.
>
> | dataset | Initial Categories |  w/o Enhancement | w Enhancement |
> | :---: | :---: | :---: | :---: |
> | COCO | 1 - 40 | 45.1 62.1 56.4   | 48.9 69.4 63.2 |
> | VOC | 1 - 10 | 77.6 89.2 85.4  | 81.5 94.4 88.2 |
>
>
> The Categorical Adaptive Label Selector **enhances the quality of the pseudo-label from the perspective of aligning overall distribution**. By fitting the distributions of positive and negative samples within each category, it dynamically selects high-quality pseudo-labels, increasing the proportion of high-quality pseudo-labels during CL. The table below presents the distribution of pseudo-label quality. **The selector significantly increased the proportion of high-quality labels**.
>
> | Method | $IoU \in [0,0.5]$ | $IoU \in [0.5,0.8]$ | $IoU \in [0.8,1.0]$ |
> | --- | :---: | :---: | :---: |
> | w/o Selector | 64.7% | 26.1% | 9.2% |
> | w Selector | 24.5% | 53.9% | 21.6% |
>
>
> Confidence Score Calibration Supervision **enhances the effectiveness of pseudo-labels in terms of supervision**. By mapping the confidence scores through a non-linear function, it better reflects the localization quality of pseudo-labels, strengthening the model's focus on high-quality labels, and enabling the model to learn from low-quality pseudo-labels. Mentioned in Figures 4(a)(b) of our paper, the nonlinear mapping function transforms the confidence scores of pseudo-labels into more favorable distributions, where higher-quality labels provide stronger supervision to the model.

---

> ### Author Response · Authors · 2024-11-20
> **Responses to Official Review by Reviewer aL8R: Part[2/4]**
>
> > **W2: About the memory and time overhead.**
>
> PseDet does not consume any additional GPU memory, nor does it require excessive disk storage or incur excessive time overhead.
>
> **(1)** GPU Memory: No additional GPU Memory is needed. The current methods, such as ERD[ref1], require loading two models during each training iteration: one is the original model used to infer the logistic of a sample, and the other is the model from the current new stage. **Our method generates pseudo labels in an offline manner.** This means that during training, **our method is the same as the regular training and does not require any additional GPU memory.** For example, based on GFL, our method occupies approximately 4.5 GB of GPU memory per GPU, whereas ERD requires more than 10 GB per GPU, nearly double the GPU memory compared to our method.
>
> **(2)** Time Efficiency: Additional time cost is negligible. Diffusion-based methods[ref2] require additional time to train a diffusion model and generate new samples, while distillation-based methods[ref1] necessitate an extra inference step in each training iteration. In contrast, our approach infers pseudo labels once in an offline manner. **This pseudo-label set can then be repeatedly used in subsequent training sessions. If training is conducted only once, the additional time overhead for this offline inference is approximately limited to 5%.**  As the number of training steps increases, the time efficiency advantage of our method becomes even more significant.
>
> **(3)** Disk Usage: The additional disk usage required by our method is negligible. After inferring pseudo labels with the previous weights, the labels can be stored. It takes only a few bytes per sample. **For the COCO dataset, the additional pseudo label .pkl files occupy approximately 10 MB of disk space.**
>
> | Method | Core | GPU Memory | Time Efficiency | Disk Usage |
> | --- | --- | --- | --- | --- |
> | ERD | KD-Based | Two Models loaded | An additional inference for one training iteration | last step checkpoint. |
> | SDDGR | Diffusion-based | Normal Train (One Model) | Additional time to generate new samples with diffusion model. | Additional samples generated by diffusion model. |
> | PseDet(Ours) | Pseudo-Label | Normal Train (One Model) | Additional time to inference Pseudo-Labels. | Every step checkpoint but only a few hundred megabytes are required for each. |
>
> In summary, our method allows pseudo-labels to be inferred in an offline manner and stored on a disk, which avoids additional resource overhead. PseDet offers advantages in terms of time efficiency and conservation of computational resources.

---

> ### Author Response · Authors · 2024-11-20
> **Responses to Official Review by Reviewer aL8R: Part[3/4]**
>
> > **Q1: About the implementation of the equation 2.**
>
> Our explanation of the implementation of Equation 2 is as follows，and these have also been added to the manuscript.
>
> Define an augmentation transformation set $A$, which encompasses transformations $A_i$ applied to image tensors. $A$ includes horizontal flip and scaling as spatial transformations. For each image, the pipeline processes the image through these transformations and inputs them into the model for inference, resulting in multiple sets of outputs for a single image.
>
> **Note that all these augmentation operators are invertible.** For inputs that are horizontally flipped, $A^{-1}$ re-flips the model's output horizontally; for scaled inputs, it scales them back to their original size. **Finally, the predictions for the same image can be fused with NMS for the final integration, as they have been inversely transformed back to their original space.**
>
> Scaling allows the model to have a more flexible perspective, as the size of objects in an image is crucial for model predictions. Reducing the size of an image helps in recognizing objects that occupy a larger area in the picture, while enlarging an image assists in identifying smaller objects within the picture. Similarly, the usage of horizontal and vertical flips serves to enhance the model's ability to recognize changes in symmetry in the left-right and top-bottom directions of images.

---

> ### Author Response · Authors · 2024-11-20
> **Responses to Official Review by Reviewer aL8R: Part[4/4]**
>
> > **Q2: Apply our method to other tasks.**
>
> Thanks for valuable suggestions, we have endeavored to apply our method to segmentation tasks and have achieved promising results. We will include them as part of our revised paper. The source code is released in [anonymous code repository](https://anonymous.4open.science/r/PseDet-F1CB). Below are the detailed experimental results and analysis:
>
> **(1) Implementation Details.** We implemented our method based on **SegFormer**[ref3] and the basic settings follow the official implementation. We selected the **ADE20K**[ref4] as our experimental dataset. This dataset presents a significant challenge due to its extensive category count of 150. It offers significant reference value for the implementation of class incremental learning.
>
> **(2) Experimental Setup.** In our experiment, we design two scenarios: **100+50 as the one-step setting** and **100+25+25 as the multi-step setting**.  The evaluation metrics include mIoU (higher is better); absolute gap (AbsGap, lower is better) and relative gap (RelGap, lower is better) between final mIoU of incremental learning and mIoU of upper-bound; forgetting percentage points (FPP, lower is better), which is used to evaluate the degree of forgetting for trained categories.
>
> **(3) Results:** The following table shows Incremental results (mIoU, %) under the one-step setting. In the first step, normal training is conducted with 100 categories. And in the second step, 50 new categories are added.
>
> | Method | Overall  | 1 - 100  | 101 - 150 | AbsGap$\downarrow$ | RelGap$\downarrow$ | FPP$\downarrow$ |
> | --- | :---: | :---: | :---: | :---: | :---: | :---: |
> | Upper Bound | 37.41 | 41.23 | 29.77 | - | - | - |
> | Catastrophic Forgetting | 0.88 | 0 | 2.63 | 36.53 | 97.64% | 43.11 |
> | Ours | **36.18** | **41.43** | **25.69** | **1.23** | **3.29%** | **1.68** |
>
>
> The following table shows the performance (mIoU, %) of our method in the more challenging multi-step scenarios of 100 + 25 + 25 scenarios.
>
> | Method | Overall | 1 - 100 | 100 - 125 | 125 - 150 | AbsGap$\downarrow$ | RelGap$\downarrow$ | FPP$\downarrow$ |
> | --- | :---: | --- | --- | --- | :---: | :---: | :---: |
> | Upper Bound | 37.41 | 41.23 | 32.94 | 26.60 | - | - | - |
> | Catastrophic Forgetting | 0.23 | 0 | 0 | 1.36 | 37.18 | 99.38% | 43.11 |
> | Ours | **35.00** | **40.63** | **25.44** | **22.04** | **2.41** | **6.44%** | **0.60** |
>
>
> Considering that segmentation tasks involve dense predictions and that, for SegFormer, there are no classification and regression branches as seen in detection tasks, it is unnecessary to consider decoupling classification and regression tasks within the context of continual learning. Consequently, pseudo-label-based methods may yield better performance when applied to segmentation tasks. Consistent with the experimental results, the model trained through continual learning exhibits an **absolute gap of only 1.23 mIoU in the one-step scenario and 2.41 mIoU in the multi-step scenario** between our method and the upper bound. **FPP gets 1.68 mIoU and 0.60 mIoU in the one-step and multi-step settings respectively**, reflecting the excellent resistance to forgetting of our method.
>
> The following table illustrates the performance (mIoU, %) in a multi-step scenario step by step. The upper bound is evaluated using the checkpoints from the official release repository. Catastrophic Forgetting occurs when finetuning is performed on new category data with the weights from the previous step. We present our results separately on three evaluation subsets.
>
> | Method | step | 1-100 | 101-125 | 126-150 | overall |
> | --- | :---: | :---: | :---: | :---: | :---: |
> | Upperbound | - | 41.23 | 32.94 | 26.60 | 37.41 |
> | Catastrophic Forgetting | step 1 | 43.11 | - | - | 43.11 |
> | | step 2 | 0 | 9.7 | - | 1.94 |
> | | step 3 | 0 | 0 | 1.36 | 0.23 |
> | Ours | step 1 | 43.11 | - | - | 43.11 |
> | | step 2 | 41.23 | 28.14 | - | 38.61 |
> | | step 3 | 40.63 | 25.44 | 22.0 | 35.00 |
>
>
> Analyzing the model's forgetting of knowledge learned at each stage from each column, for the initial 1-100 categories, the model's initial performance exceeds that of the upperbound. However, as the model capacity is reallocated, the evaluation performance for the first 100 categories declines. In the final stage, our method shows a significant gap compared to fine-tuning, consistent with the performance in detection tasks. **Our method transfers well to segmentation tasks and performs effectively in multi-step scenarios**.
>
> > **References:**
> > [ref1] Generalized Focal Loss: Learning Qualified and Distributed Bounding Boxes for Dense Object Detection. NeurIPS 2020
> > [ref2] SDDGR: Stable Diffusion-based Deep Generative Replay for Class Incremental Object Detection. CVPR 2024
> > [ref3] SegFormer: Simple and Efficient Design for Semantic Segmentation with Transformers. NeurIPS 2021
> > [ref4] Scene Parsing Through ADE20K Dataset. CVPR2017

---

### Meta-Review · Area_Chair_J1Qk · 2024-12-15

**Metareview:**

This paper introduces a novel framework addressing key limitations in pseudo-labeling for incremental object detection (IOD). The paper identifies three critical challenges: noisy pseudo-labels, fixed thresholding for label filtering, and misaligned confidence scores. The proposed framework, PseDet, tackles these through a Spatio-Temporal Enhancement module, a Categorical Adaptive Label Selector, and Confidence Score Calibration. These components collectively aim to enhance pseudo-label quality and alignment, thereby reducing catastrophic forgetting and improving detection performance. The work demonstrates state-of-the-art results on the COCO dataset across various settings.

During the author-reviewer discussion period, reviewers raised concerns about the novelty of the core idea, the training time and memory costs of the Spatio-Temporal Enhancement module, etc. The authors responded by providing additional results. They clarified implementation details for Equation 2 and addressed memory concerns by highlighting the method's offline pseudo-label generation, which avoids excessive resource overhead. They also extended their approach to segmentation tasks, showing its generalizability. After the rebuttal period, two of the three reviewers are positive towards this paper. Reviewer aL8R still has remaining concerns about the memory assumption, which seems impractical and unrealistic in many scenarios.

Given that most reviewers are positive about the paper and there is no strong objection to accepting it, the paper is accepted. The authors are required to include the additional results and discussions in the final version. Besides, they should also discuss the limitations raised by Reviewer aL8R in their camera-ready version.

**Additional Comments On Reviewer Discussion:**

During the author-reviewer discussion period, reviewers raised concerns about the novelty of the core idea, the training time and memory costs of the Spatio-Temporal Enhancement module, etc. The authors responded by providing additional results. They clarified implementation details for Equation 2 and addressed memory concerns by highlighting the method's offline pseudo-label generation, which avoids excessive resource overhead. They also extended their approach to segmentation tasks, showing its generalizability. After the rebuttal period, two of the three reviewers are positive towards this paper. Reviewer aL8R still has remaining concerns about the memory assumption, which seems impractical and unrealistic in many scenarios.

Given that most reviewers are positive about the paper and there is no strong objection to accepting it, the paper is accepted. The authors are required to include the additional results and discussions in the final version. Besides, they should also discuss the limitations raised by Reviewer aL8R in their camera-ready version.

---

### Decision · Program_Chairs · 2025-01-22

Accept (Poster)